# Ungulate presence and predation risks reduce acorn predation by mice in dehesas

**Teresa Morán-López**[ID][1], **Jesús Sánchez-Dávila**[2], **Ignasi Torre**[3], **Alvaro Navarro-Castilla**[ID][4], **Isabel Barja**[4,5], **Mario Díaz**[ID][2]*

**1** Laboratorio Ecotono, INIBIOMA-CONICET, Universidad Nacional del Comahue, Black River, Argentina, **2** Department of Biogeography and Global Change (BGC-MNCN-CSIC), National Museum of Natural Sciences, CSIC, Madrid, Spain, **3** Museu de Ciències Naturals de Granollers (MCNG), Granollers, Barcelona, Spain, **4** Departament of Biology, Unit of Zoology, Faculty of Sciences, Universidad Autónoma de Madrid, Madrid, Spain, **5** Centro de Investigación en Biodiversidad y Cambio Global (CIBC-UAM) Universidad Autónoma de Madrid, Madrid, Spain

* Mario.Diaz@ccma.csic.es

**Data Availability Statement:** All relevant data are within the paper and its Supporting Information files.

## Abstract

Foraging decisions by rodents are key for the long-term maintenance of oak populations in which avian seed dispersers are absent or inefficient. Decisions are determined by the environmental setting in which acorn-rodent encounters occur. In particular, seed value, competition and predation risks have been found to modify rodent foraging decisions in forest and human-modified habitats. Nonetheless, there is little information about their joint effects on rodent behavior, and hence, local acorn dispersal (or predation). In this work, we manipulate and model the mouse-oak interaction in a Spanish dehesa, an anthropogenic savanna system in which nearby areas can show contrasting levels of ungulate densities and antipredatory cover. First, we conducted a large-scale cafeteria field experiment, where we modified ungulate presence and predation risk, and followed mouse foraging decisions under contrasting levels of moonlight and acorn availability. Then, we estimated the net effects of competition and risk by means of a transition probability model that simulated mouse foraging decisions. Our results show that mice are able to adapt their foraging decisions to the environmental context, affecting initial fates of handled acorns. Under high predation risks mice foraged opportunistically carrying away large and small seeds, whereas under safe conditions large acorns tended to be predated *in situ*. In addition, in the presence of ungulates lack of antipredatory cover around trees reduced mice activity outside tree canopies, and hence, large acorns had a higher probability of survival. Overall, our results point out that inter-specific interactions preventing efficient foraging by scatter-hoarders can reduce acorn predation. This suggests that the maintenance of the full set of seed consumers as well as top predators in dehesas may be key for promoting local dispersal.

## Introduction

Scatter-hoarders are key seed dispersers in temperate and Mediterranean forests dominated by oaks [1, 2]. Nut dispersal by scatter-hoarders (synzoochory) is a classical plant-animal

**Funding:** MD. Grant number S2013/MAE-2719. REMEDINAL3-CM project, funded by the Autonomous Community of Madrid. NO.

conditional mutualism. The outcome of the interaction may be either mutualistic (dispersal) or antagonistic (predation) depending on the proportion of seeds consumed *vs.* cached and not retrieved [3]. The balance between mutualism and antagonism is contingent on intrinsic properties of interaction partners (e.g. propensity of animals to store food) as well as on the ecological setting in which the interaction occurs [2]. As a result, the net effects of synzoochory can be highly dynamic in space and time, making it difficult to predict its outcomes along environmental gradients and ecological timescales [4, 5].

Acorn dispersal depends on scatter-hoarder corvids and rodents. Corvids disperse acorns tens to hundreds of meters [6], whereas rodents transport acorns locally and a high proportion of them are eventually predated [7, 8]. Nonetheless, several mouse species (*Apodemus*, *Mus*, *Peromyscus*) can become important acorn dispersers in landscapes where scatter-hoarding corvids are absent [9] or become inefficient [10]. Two main external factors modulate mouse foraging decisions: competition for seeds and predation risk [11–13]. Intraspecific competition and the presence of ungulates tend to encourage seed dispersal [3, 14–16]. Especially, when predating seeds *in situ* is more time-consuming than carrying them away, and hence, scatter-hoarding facilitates stockpiling seeds before they are depleted by competitors [12, 17]. The effects of risk perception on mouse foraging decisions depend on factors that affect exposure to predators (e.g. moonlight) as well as direct cues of their presence (e.g. scent) [18–23]. In general, intermediate risks can promote acorn removal over predation when mice carry away seeds to manipulate them in safer locations or when handling times of consuming seeds *in situ* are too long [12]. However, if lack of cover in the vicinity of trees triggers predation risks, acorn mobilization distances and caching rates can be significantly reduced [13, 16]. In general, suboptimal conditions for foraging mice (i.e. competition and predation risk) tend to favor scatter-hoarding over *in situ* predation. In the absence of stress, rodents usually act as efficient seed predators consuming, immediately or soon afterwards, seed crops under the canopy of mother trees [2].

Beyond the environmental conditions of plant-animal encounters, seed size can affect the initial outcomes of the interaction (selected, eaten or cached) as well as post-dispersal processes such as germination and seedling survival. Larger seeds are usually selected and preferentially cached because they provide higher food rewards [7, 24–28]. In addition, seed size enhances post-dispersal seedling survival and establishment [29], which is a key component of dispersal effectiveness [30] in scatter-hoarder animals [31]. Nonetheless, the strength and even sign of acorn size effects on mouse foraging decisions are not unequivocal, but context-dependent. Larger acorns are most preferred when food is scarce [32–34], but may be avoided when longer handling times [25] diminish their profitability [35, 36] or result in unaffordable predation risks during manipulation [11, 12]. Therefore, to have a full picture of mice role in acorn dispersal we need to account for seed size effects on scatter-hoarding decisions as well as the influence of competition and risk.

In this context, dehesas represent an excellent study system to assess the main factors modulating mouse foraging decisions, and hence, acorn dispersal. They are savanna-like habitats, simpler than natural forests but diverse enough to maintain all key elements influencing the mouse-oak conditional mutualism. Depending on the local intensity of management, nearby areas can have contrasting levels of shrub cover and competition with ungulates [18, 37]. In addition, the community of predators is simpler than in forested areas, facilitating the experimental manipulation of direct cues of risks [23]. In this work we take advantage of a large-scale experiment of ungulate exclosure in a Mediterranean dehesa to (1) quantify acorn size effects across different stages of the dispersal process (from seed selection to initial fates after transportation); and (2) evaluate if size effects are consistent across contrasting scenarios of predation risk and competition for seeds. In addition, we parameterized a transition

probability model that assembled all scatter-hoarding decisions by mice to quantify and tease apart the net effect of competition and risk on acorn dispersal. We expected that suboptimal conditions for mice (i.e. competition and risk) would constrain their ability to forage efficiently, thus reducing acorn predation.

## Methods

### Study area and species

Field work was carried out in the holm oak *Quercus ilex* dehesa woodlands of the Cabañeros National Park (Central Spain, Ciudad Real province, 39˚24' N, 38˚35 W). Dehesas are savanna-like man-made habitats resulting from shrub removal and tree thinning and pruning to enhance herb growth for livestock [38]. The studied dehesas were opened through tree thinning from the original Mediterranean forests in the late 1950s. Currently they have no livestock but wild ungulate populations of red deer *Cervus elaphus* and wild boars *Sus scrofa*. Deer densities were around 0.14 ind./ha [39] and boars are abundant but at unknown densities [40]. Acorns fall from trees from mid-October to late November [9].

The study area covers around 780 ha, with two sites separated by 1500 m. Average tree density in the area is 20.4 trees ha-1, although it is higher at site 1 (30.05 and 7.4 trees per ha at site 1 and 2, respectively). Tussocks and grasses are the main vegetation cover around trees (94.1%, on average), whereas shrub cover is low (<1%) [23]. At each site there is an ungulate exclosure (150 and 4.65 ha, site 1 and 2 respectively) made with wire fences 2 m tall and 32 cm x 16 cm mesh. The exclosures prevent the entrance of ungulates but not of mesocarnivores (mainly common genets *Genetta genetta* and red foxes *Vulpes vulpes*; pers. obs. based on scat searches) and raptors. In addition, ungulate exclosures have modified the structure of the vegetation. Lack of ungulate browsing has resulted in taller vegetation around trees (21.3±1.3 vs 8.3±0.7 cm inside and outside exclosures, respectively) and higher covers of taller resprouts under canopies (30.1±3.4 vs 19.3±2.5% and 79.9±13.4 vs 23.2±3.1 cm, respectively [41]). To evaluate the effects of the presence of ungulates on mouse foraging decisions at each site we established half of focal trees outside the exclosure and half of them inside it (see below). At site 1 we worked in the southernmost 5.72 ha of the site 1 exclosure and in the whole 4.65 ha site 2 exclosure, both paired with a close-by similar area outside the exclosure.

### Experimental design

Tree occupancy by mice was established by means of live trapping using Sherman traps (23 × 7.5 × 9 cm; Sherman Co., Tallahassee, USA) baited with canned tuna in olive oil mixed with flour and a piece of apple. Water-repellent cotton was provided to prevent the cooling of the individual captured overnight. Traps were set during two consecutive days during the new moon of January 2012. High capture probability of *M. spretus* (detectability: 0.88±0.03 SE; [42]) pointed out that false negatives in occupancy was unlikely. Among trees known to be occupied by Algerian mice, we randomly selected ten trees inside and ten outside in each of the two exclosures (40 focal trees in total).

We paired focal trees according to their proximity and we randomly assigned a predator scent treatment to one of them. Predator scent treatment consisted of placing fresh genet feces (10 g) mixed with distilled water close to a corner of the cages where acorns were placed [23]. Genets are generalist predators whose presence and scats are known to influence rodent behavior [20, 22, 43]. Fresh feces were collected from two captive common genets housed in the Cañada Real Open Center (Madrid, Spain).

Fresh acorns were collected from holm oaks growing near the study area in October 2011 and stored dry in a cooler (4˚C) until use. Sound acorns, with no marks of insect damage [44],

were weighed with a digital balance to the nearest 0.01 g. To offer a full range of acorn sizes, in each cafeteria trial (combination of tree, month and moon light, see below) we randomly selected 5 large (>10 g), 5 medium-sized (5–10 g) and 5 small (1–5 g) acorns. Acorns were placed under the canopy of each focal tree inside a 50 cm × 50 cm x 15 cm galvanized-steel cage to prevent acorn consumption by birds or ungulates [44]. Cages were located 1.2 m on average (range 0.3–2.7 m) from focal tree trunks. A metal wire (ø 0.6 mm, 0.5 m length) with a numbered plastic tag was attached to each acorn [45]. After removing any naturally-present acorns within the cages, we randomly placed acorns in the intersection of a 3 row x 5 column grid. To track mouse choices, acorn size for each position was noted. Acorns were left exposed to mice for three consecutive nights, then removed. Acorns carried away from cages were searched by looking for plastic tags in circles around focal trees (up to 30 m away, where most acorns are initially deposited [27, 46]). Searches were performed during the following days of acorn offering (24 and 72 hours). We tracked the status of acorns that were transported and not predated throughout the experiment. We considered an acorn to be predated when it was either completely consumed (only wire and tag was found, sometimes with husk remains attached) or partially consumed in its apical side thus removing the embryo. To account for changes in night brightness and acorn availability [21, 47], the cafeteria experiment was repeated four times during the full-moon and new-moon periods of November 2011 and February 2012. No official permits or protocol approvals were legally necessary since we did not manipulate individual mice except for checking whether trees were occupied or not by means of live traps. We followed Guidelines of the American Society of Mammalogists for the use of wild mammals in research [48]. We performed all manipulations with disposable latex gloves, to avoid effects of human odor on rodent behavior [49].

## Mouse foraging behavior

A video-camera OmniVision CMOS 380 LTV (OmniVision, Santa Clara, USA) (3.6 mm lens) monitored mouse foraging activity within each cage. Cameras were set on 1.5 m tall tripods located 2.5 m from each cage, powered by car batteries (70 Ah, lead acid) connected to a solar panel (ono-silicon erial P_20; 20 w). Video-cameras were connected to ELRO recorders with dvr32cards (ELRO, Amsterdam, Netherlands) and took continuous recording for three consecutive days autonomously (recorded in quality at 5 frames s$^{-1}$). Events with rodent activity, from the entry of the individual into the cage up to the exit from it, were located and separated using Boilsoft Video Splitter software (https://www.boilsoft.com/videosplitter/) [43]. Within each foraging event, we noted which acorn was handled (selection) and if it was removed outside the cage or not. For removed acorns we measured transportation distances (cm) and noted its status (predated or not after transportation).

## Data analysis

To assess acorn selection by rodents, we fitted a hierarchical multinomial model. For each foraging event, we modeled which acorn was handled (out of those available in the cage) as a function of acorn size (g), moon phase (new/full), month (February, November), ungulate presence (yes/no), predator scent (yes/no), acorn availability in the cage (g) and the two-way interactions between size and environmental effects. Local acorn availability was measured as total acorn mass in the cage during the event. Both acorn size and availability were scaled previous to the analyses (mean = 0, SD = 1) so that we could compare the magnitude of covariate effects. Focal tree was introduced as a random factor in the intercept term to account for repeated sampling during the experiment. Subsequently, we evaluated the effects of acorn size, competition and risk on the foraging decision of carrying acorns outside the cage or not

(acorn removal, hereafter). To this end we used a hierarchical logistic model. Our response variable was acorn removal (yes/no). Our explanatory variables and random effects were the same as in the multinomial model.

Finally, we analyzed the effect of acorn size and environmental covariates (and their two-way interaction) on seed removal distances and initial fates. Our response variables were transportation distances of acorns (cm, log-transformed) and deposition status (viable or predated). We used a hierarchical Gaussian model in the former case, and a hierarchical logistic model in the latter. Our explanatory variables and random effects were the same as in the previous models. In all four models (acorn selection, removal, transportation distance and deposition fate) we used uninformative priors (S1 File). All analyses were performed employing a Bayesian approach with JAGS 3.4.0 [50]. We checked for convergence for all model parameters (Rhat < 1.1) and that the effective sample size of posterior distributions was high (>800). We estimated the mean and credible interval of posterior distributions, calculated the proportion of the posterior distribution with the same sign of the mean (f) and evaluated the predictive power of our models by means of posterior predictive checks (S1 and S2 Files).

## Simulating scatter-hoarding decisions

To estimate the joint effect of seed size, competition and risk on acorn dispersal we designed a probability transition model in which simulated mice adapted their foraging behavior to the environmental context (S3 File). Before model run, we parameterized mouse scatter-hoarding decisions (from acorn selection to initial fate of transported acorns) following the same scheme of regressions explained in the previous section. Here, we only used data from November, the period of peak acorn falling in our study system. Thus, we did not include month as a covariate. For each behavioral submodel (selection, removal and initial fate), we obtained posterior distributions of parameters by running 50000 iterations in three chains (in all cases Rhat< 1.1, and Neff> 1000).

Model setup mimics our experimental design, 20 trees outside and 20 inside exclosures paired according to a predator scent treatment (presence vs. absence). Simulations begin under new moon conditions with focal trees offering 15 acorns of large, medium and small sizes (5 each). Acorn size is sampled from empirical distributions of these size categories. In each focal tree, the number of foraging events is drawn from a Poison distribution with mean equal to the average number of events observed in the corresponding moon phase. During each foraging event, simulated mice decide which acorn to handle and whether to remove it or not. If removed, mice decide to predate it or not after mobilization and acorn availability in the cage is updated. Once all foraging events (of all trees) are simulated, acorn dispersal is modelled under full moon conditions (S1 Fig in S3 File).

For each model run we sampled parameter of behavioral submodels (selection, removal and deposition) from posterior distributions fitted to data. Thus, in our simulations, mice adapted their decisions to acorn size and availability (in the experimental cage), characteristics of the focal tree (i.e. ungulate and predator scent presence), and the moon phase in which the foraging event occurs (new or full moon). After each model run (simulated mice foraging under new and full moon conditions), the program tracked the size and status of handled acorns and the environmental covariates in which the foraging event occurred. We run the model 1000 times and plotted deposition rates of viable acorns and their size with respect to the moon phase and tree characteristics (predator scent and ungulate presence). See S3 File for detailed model specifications and S1 Fig in S3 File for a summary of the process overview.

## Results

Before setting the cafeteria experiments in November, we removed from cages 53.3 acorns/m$^2$ on average (range: 0–104). No acorns were found in February. We detected *Mus spretus* activity in 18 and 26 trees in the new and full moon of November; and in 26 and 24 trees in February. Therefore, we finally monitored 1410 acorns instead of 2400 (40 initial focal trees x 15 acorns per trial x 2 months x 2 moonlight conditions). Mice (*M. spretus*) handled 986 acorns (69.5% of those offered). Out of them, 288 (29.2%) were carried outside cages and 211 (73.2%) were relocated, 67 of which (31.8%) were not predated after transportation, and 8 out of these (11.9%) were found buried.

### Foraging decisions in the focal tree: Selection and removal

In general, mice preferentially handled larger acorns, but the positive effect of size was modulated by environmental conditions. Size-driven selection occurred in the absence of competition with ungulates (Fig 1A) and predator scent (Fig 1B). In addition, mouse selectivity was enhanced under low local acorn availability (Table 1, selection). Among handled acorns, mice preferentially removed smaller ones outside the cages. Such behavior occurred when risks were low due to reduced night brightness (new moon, Fig 1C) or lack of predator scent (Fig 1D), as well as when ungulates were absent (Table 2). Acorn availability at local and landscape

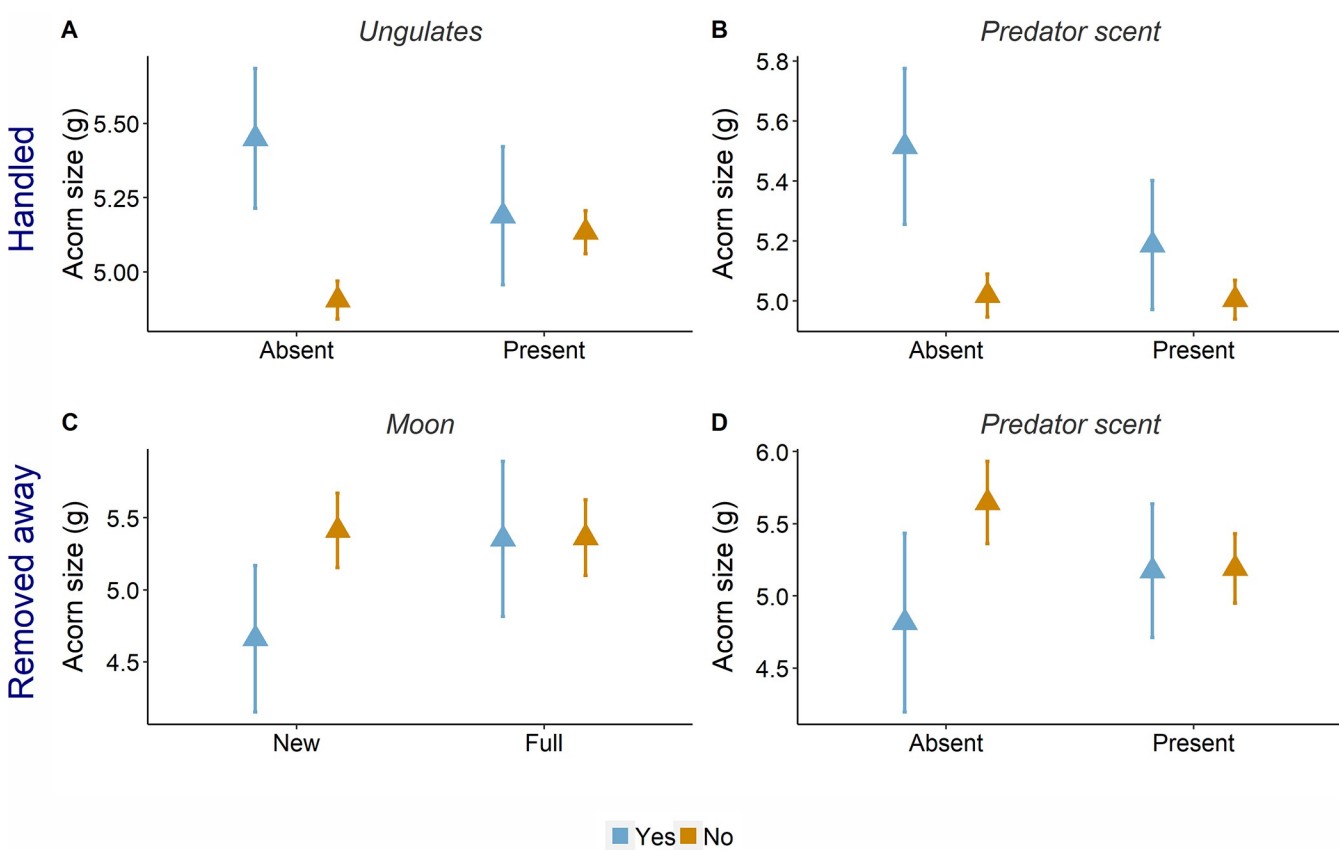

**Fig 1. Mouse foraging decisions during acorn selection and removal (upper and lower panels, respectively).** Size of handled acorns in the presence or absence of (A) ungulates and (B) predator scent. Size of acorns (removed away from the cage or not) (C) under new or full moon conditions and (D) in the presence or absence of predator scent. Point colors depict whether the acorn was selected or removed (yes, blue) or not (no, orange). In all cases acorn size is expressed in grams. Points represent mean values, bars standard errors (N = 1677 foraging events).

**Table 1. Summary table of the effects of size, moonlight, month, ungulate presence, predator scent and local acorn availability (and their interactions with size) on the probability of acorn selection and removal.** A total of 1677 foraging events were analyzed.

| Process | Fixed effect | Mean | HPD1 | f | |
|---|---|---|---|---|---|
| **Acorn selection** | **Size** | **0.19** | **[0.09, 0.29]** | **1.00** | ** |
| | Moon (Full) | 0.03 | [-6.22, 6.35] | 0.50 | |
| | Month (February) | -0.03 | [-6.12, 6.06] | 0.50 | |
| | Ungulate (Yes) | -0.05 | [-6.27, 6.16] | 0.51 | |
| | Scent (Yes) | 0.01 | [-6.16, 6.28] | 0.50 | |
| | Availability | -0.02 | [-6.32, 6.19] | 0.50 | |
| | Size*Moon | 0.07 | [-0.03, 0.17] | 0.93 | |
| | Size*Month | -0.06 | [-0.16, 0.04] | 0.88 | |
| | **Size*Ungulates** | **-0.13** | **[-0.23, -0.03]** | **0.99** | ** |
| | **Size*Scent** | **-0.08** | **[-0.18, 0.01]** | **0.96** | * |
| | **Size*Availability** | **-0.04** | **[-0.09, 0.01]** | **0.95** | * |
| **Acorn removal** | **Size** | **-0.50** | **[-0.94, -0.07]** | **0.99** | ** |
| | Moon (Full) | 0.07 | [-0.27, 0.39] | 0.65 | |
| | **Month (February)** | **0.77** | **[0.43, 1.11]** | **1.00** | ** |
| | Ungulate (Yes) | -0.22 | [-0.96, 0.47] | 0.73 | |
| | Scent (Yes) | 0.20 | [-0.53, 0.93] | 0.72 | |
| | **Availability** | **0.29** | **[0.12, 0.46]** | **1.00** | ** |
| | **Size*Moon** | **0.29** | **[-0.02, 0.60]** | **0.96** | * |
| | Size*Month | -0.09 | [-0.41, 0.22] | 0.71 | |
| | **Size*Ungulates** | **0.24** | **[-0.07, 0.55]** | **0.94** | · |
| | **Size*Scent** | **0.30** | **[0.00, 0.59]** | **0.98** | * |
| | Size*Availability | 0.10 | [-0.07, 0.26] | 0.88 | |

Mean of posterior distribution, highest posterior density interval (HPD) and percentage of the posterior distribution with the same sign as the mean (f) are shown. Effects with f ≥ 0.95 are in bold. • depicts f ∈ [0.90, 0.95)

scales did not modify size effects, although they changed removal rates. They were lower during the acorn fall peak (13% in November *vs* 24% in February), whereas local acorn availability (i.e. in cages) enhanced removal (Table 2, removal).

## Foraging decisions after removal: Distances and predation after deposition

Mice transported acorns shorter distances under new moon conditions (Fig 2A) and when ungulates were present (Fig 2B). During lean periods (February) transportation distances and post-dispersal predation increased (Table 2, Month). In addition, larger acorns were preferentially predated (Fig 2C), though the presence of ungulates and full moon conditions attenuated this negative effect (Fig 2D, Table 2). Regarding the microhabitat of deposition, viable acorns were frequently found under tree canopies or close to oak resprouts (96.8% and 97.9% of acorns, inside and outside exclosures, respectively). Only 2.4% of transported acorns were deposited in open areas.

## Transition probability model for acorn dispersal

Under optimal conditions (new moon, no predator scent or ungulates), post-dispersal predation rates were higher (Fig 3A) and simulated mice preferentially consumed large acorns (i.e. viable acorns -blue bars- were smaller, Fig 3B–3D). In contrast, predation risks and ungulate presence precluded acorn predation after mobilization and attenuated selection. As a result, the proportion of viable acorns increased and they were larger (Fig 3A–3D).

**Table 2. Summary table of the effects of size, moonlight, month, ungulate presence, predator scent and local acorn availability (and their interactions with size) on acorn mobilization distances and the probability that it is deposited in a viable status (vs predated).** A total of 211 acorns that were mobilized outside cages and retrieved were analyzed.

| Process | Fixed effect | Mean | HPD | F | |
|---|---|---|---|---|---|
| **Mobilization distance** | Size | 0.16 | [-0.51, 0.83] | 0.68 | |
| | **Moon (Full)** | **-0.67** | **[-1.27, -0.06]** | **0.98** | * |
| | Month (February) | **0.54** | **[-0.14, 1.2]** | **0.94** | · |
| | **Ungulate (Yes)** | **-0.75** | **[-1.59, 0.14]** | **0.95** | * |
| | Scent (Yes) | 0.09 | [-0.73, 0.98] | 0.57 | |
| | Availability | -0.01 | [-0.31, 0.29] | 0.52 | |
| | Size*Moon | -0.07 | [-0.66, 0.49] | 0.60 | |
| | Size*Month | -0.33 | [-0.94, 0.28] | 0.86 | |
| | Size*Ungulates | 0.18 | [-0.44, 0.81] | 0.71 | |
| | Size*Scent | 0.22 | [-0.36, 0.79] | 0.78 | |
| | Size*Availability | -0.16 | [-0.49, 0.17] | 0.83 | |
| **Viability after deposition** | **Size** | **-1.20** | **[-2.15, -0.33]** | **1** | * |
| | Moon (Full) | 0.42 | [-0.38. 1.22] | 0.85 | |
| | **Month (February)** | **-1.58** | **[-2.46, -0.75]** | **1** | * |
| | **Ungulate (Yes)** | **0.67** | **[-0.32, 1.69]** | **0.91** | · |
| | Scent (Yes) | -0.16 | [-1.14, 0.80] | 0.63 | |
| | **Availability** | **0.52** | **[0.10, 0.97]** | **0.99** | * |
| | **Size*Moon** | **0.66** | **[-0.12, 1.46]** | **0.95** | * |
| | Size*Month | 0.49 | [-0.33, 1.34] | 0.88 | |
| | **Size*Ungulates** | **0.59** | **[-0.22, 1.40]** | **0.92** | · |
| | Size*Scent | 0.21 | [-0.52, 0.94] | 0.72 | |
| | Size*Availability | -0.24 | [-0.73, 0.25] | 0.84 | |

Mean of posterior distribution, highest posterior density interval (HPD) and percentage of the posterior distribution with the same sign as the mean (f) are shown. Effects with f ≥ 0.95 are in bold. • depicts f ∈ [0.90, 0.95].

## Discussion

Overall, our work shows that mice are able to adapt their foraging decisions to the presence of ungulates and perceived predation risk, and that such behavioral adjustments affect the fate of acorns at initial stages of the dispersal process. When not exposed to stressful factors, mice preferentially consumed *in situ* large acorns and carried away small ones. Furthermore, seeds were more likely to be predated after deposition. In contrast, under stressful conditions (increased predation risk and ungulate presence) mice foraged opportunistically and reduced their activity outside tree canopies. As a result, predation rates of seeds decreased, and larger acorns had a higher probability of survival, at least in the short term. This bolsters the idea that interactions with third-party players can modify the qualitative component of dispersal effectiveness of scatter-hoarding rodents [12, 15, 51].

As expected, larger and more valuable acorns were preferentially handled by mice, which adapted this behavior to the environmental context [12]. In line with previous work, mice foraged opportunistically in trees with predator scent, probably because they devoted more time to vigilant behaviors [15, 43] at the expenses of acorn discrimination [21]. In contrast, acorn availability effects did not follow the expectations of increased selectivity in scenarios of food depletion or competition [27, 51, 52]. Seed size effects were similar between acorn fall peaks and lean periods. In addition, mice foraged randomly in the presence of ungulates, while selected larger seeds in their absence. These unexpected results may respond to some

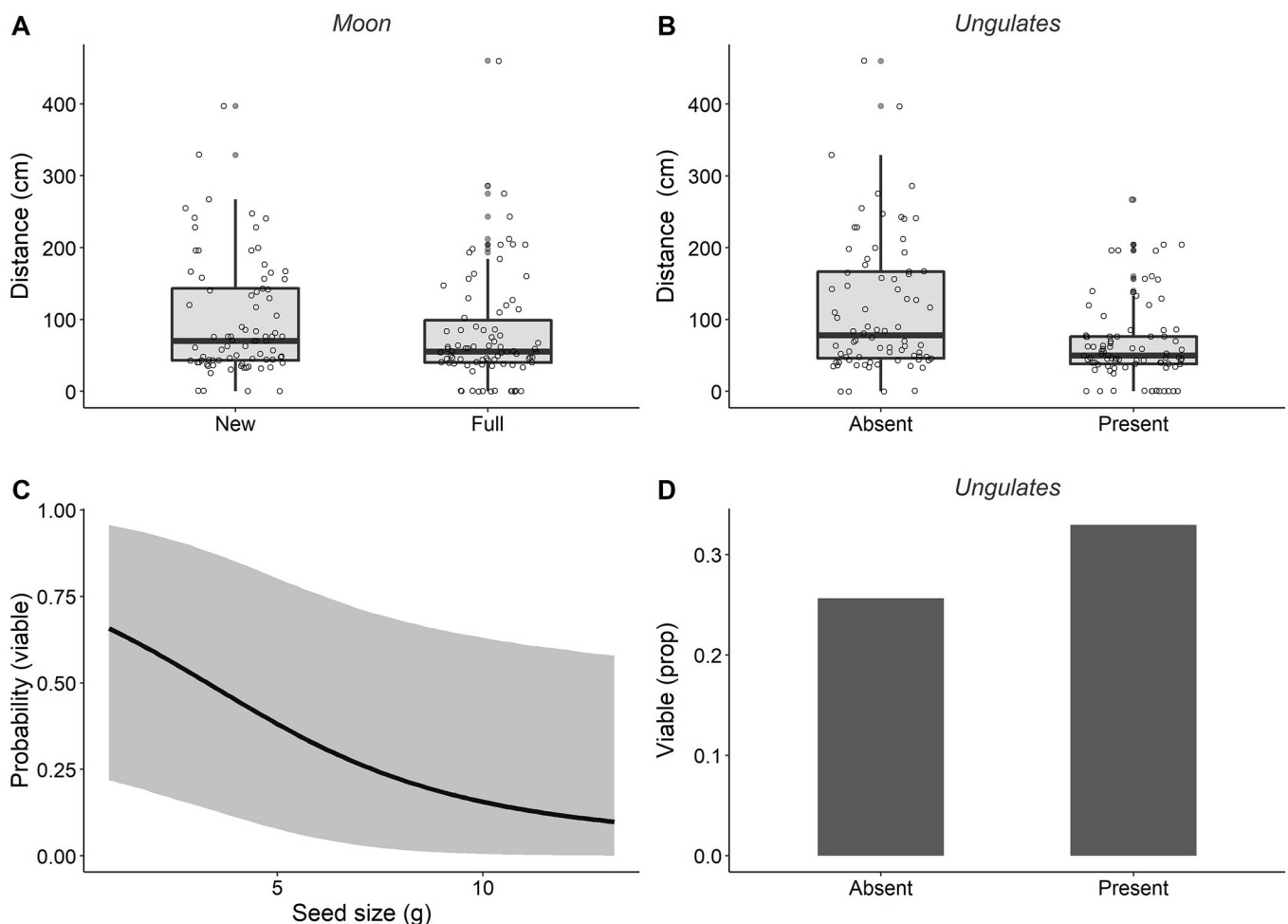

**Fig 2. Mouse foraging decisions during transportation and after deposition (upper and lower panels, respectively).** Removal distances under (A) new and full moon conditions and (B) in the absence and presence of ungulates. (C) Seed size effects on the probability of predation after deposition (black line represents mean effects and shaded area 0.95 credible intervals). (D) Proportion of acorns escaping predation after deposition in the absence and presence of ungulates. Sample size was 267 for mobilization distances and 211 for analyses of initial seed fate.

particularities of our study system. On one hand, dehesas are characterized by a high acorn production [53, 54], and hence, the effects of competition for seeds may have been attenuated [13]. On the other, ungulates not only compete with rodents for acorns, but also are important modulators of vegetation structure in dehesas. Outside exclosures, grazing and trampling by ungulates has led to scarcer and shorter vegetation around focal trees, whereas inside exclosures tall resprouts, grasses and tussocks can be found (S1 Fig in S5 File). Such changes in vegetation structure allow mice to forage under shelter, devoting less time to vigilant behaviors [43], and hence, selecting the most profitable food items. In line with previous work, our results suggest that predation risks rather than competition modulate mouse foraging decisions in dehesas [43]. Also, that the effects of ungulate presence on vegetation structure can strongly affect the foraging behavior of scatter-hoarding rodents [17].

Larger acorns tend to be carried away, transported farther and preferentially cached in forest habitats [7, 26, 55, 56]. However, in our study larger acorns had a higher probability of being predated (*in situ* and after transportation) and seed size did not affect transportation distances. Again, these results highlight that in dehesas environmental conditions are particularly

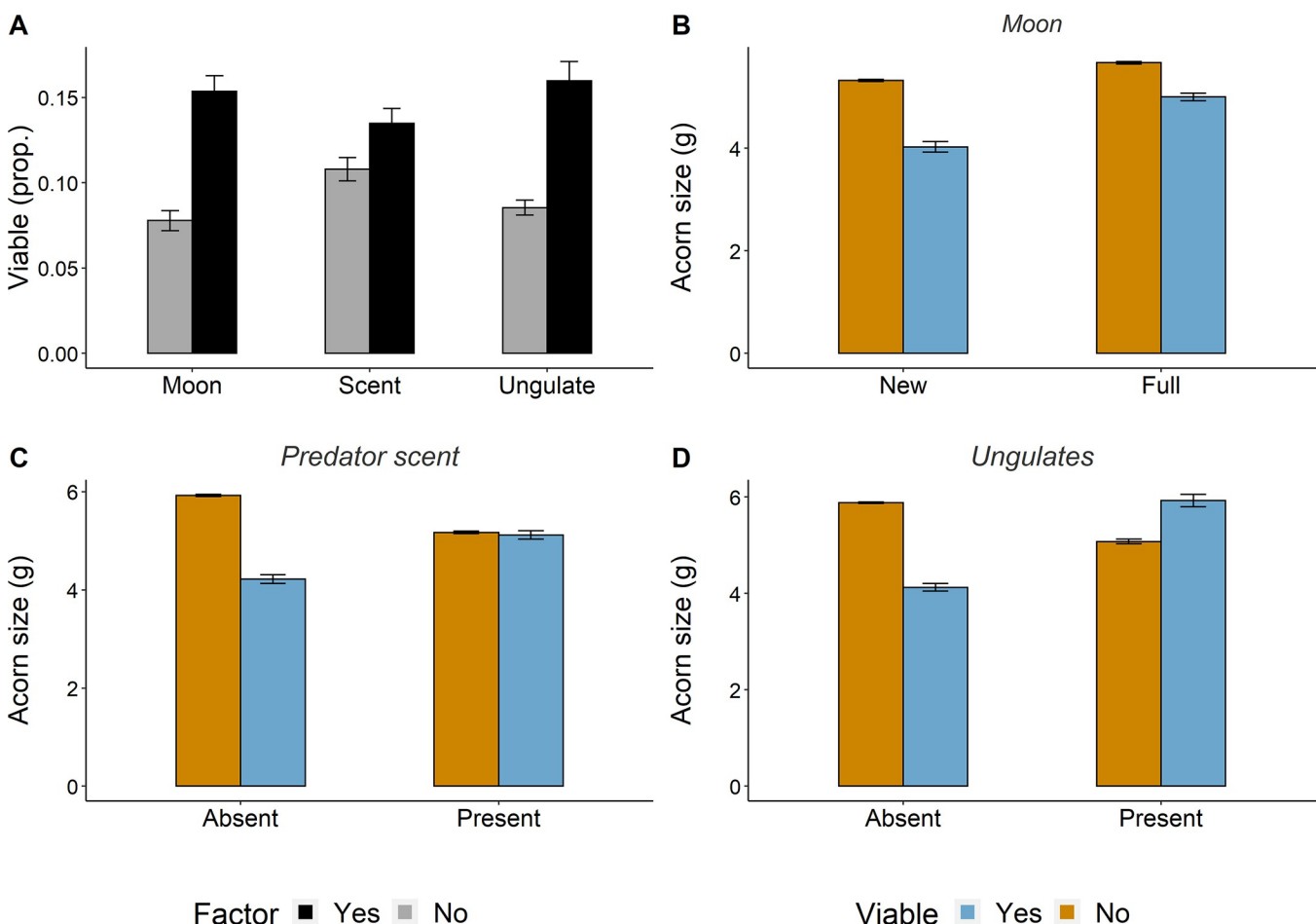

**Fig 3. Results from simulations of the probability transition model for acorn dispersal.** (A) Proportion of acorns escaping predation in the presence or absence of environmental stressors (i.e. full moon, ungulates present or predator scent, black bar) in comparison to more optimal conditions (i.e. new moon, ungulates absent, no predator scent, grey bar). Size of predated (yellow) and viable (blue) acorns under (B) new *vs* full moon conditions and in the presence or absence of (C) ungulates and (D) predator scent. Bars represent mean values (±s.e.) across 1000 simulations.

harsh for rodents. In general, rodents preferentially carry away small seeds when the costs of transporting large ones are unaffordable [12]. In the presence of ungulates, low antipredatory cover and high trampling risks may have triggered transportation costs [13, 57], deterring mice from carrying large seeds away. Seed size effects were not fixed, but depended on direct and indirect cues of risk. Preferential removal of small seeds only occurred in trees with no predator scent or under new moon conditions, reflecting that only when risks are reduced mice can take the time to select among the seeds available [15, 21].

Regarding initial seed fate, we expected higher predation rates when acorns were deposited close to tree canopies [13, 16]. Nonetheless, this relationship blurred in our system. In the presence of ungulates, larger acorns had a higher probability of escaping predation in spite of being mobilized nearby source trees. In dehesas, outside ungulate exclosures the pervasiveness of open land cover forces mice to concentrate their activities beneath canopies [13, 23, 41], and decreases the likelihood that mobilized acorns are encountered and consumed [58]. Accordingly, in our simulations, suboptimal conditions (due to increased risks or ungulate presence) discouraged mice from selecting which acorns to handle and carry away, and from consuming seeds after transportation. Consequently, predation rates were reduced and larger acorns had a

higher probability of survival. In principle, these results suggest that intermediate levels of stress can enhance the probability of acorn dispersal by rodents (as suggested by [59]). Nonetheless, a high proportion of acorns were deposited within resprouts growing under tree canopies, where the establishment of a new seedlings is highly unlikely [60]. Therefore, it remains an open question whether mice can act as local acorn dispersers in dehesas by outweighing a very low dispersal quality with high removal rates (as found in forest habitats [29]).

This work builds on previous research analyzing the effects of competition and risk on mouse foraging behavior in dehesas [43]. In the present study, by accounting for many stages of the scatter-hoarding process (from initial acorn selection to predation after transportation [57]) and including the entire acorn fall season [26] as well as contrasting moon light conditions [21], we obtained a more in-depth understanding of the role of mice as acorn dispersers (or predators) in dehesas. Overall, our results show that suboptimal conditions for mice can reduce predation rates and increase the probability that larger ones survive acorn handling. Also, they suggest that in dehesas the effects of ungulates on mouse foraging decisions are mediated by their impacts on vegetation cover rather than by competition. Nonetheless, low caching rates (<1%) prevented us from analyzing scatter-hoarding (in spite of tracking 1410 acorns). Such difficulties are commonplace in dehesas, where caching rates by mice are low (reported values lie between 1.83% [13] to 7.52% [58]). In this context, mechanistic models like ours result particularly useful [61]. They allow to simulate a high number of foraging events, and hence, monitor the fate of those that are rare but important from a demographic perspective (e.g. survival of cached acorns). However, to quantify seed dispersal effectiveness by mice (*sensu* [30]), our model needs information about caching rates and long-term survival. To obtain robust estimates of caching rates, a higher number of acorns could be tracked at the expense of simplifying the number of environmental factors being evaluated (e.g. only ungulate presence) and of not videorecording foraging events. In the case of cache survival rates, we believe that sowing acorns and monitoring artificial caches seems the only way to achieve adequate sample sizes. Although such approach does not allow to evaluate the foraging decisions made by cache owners, it informs about survival rates of cached acorns from pilferers and ungulates as well as the probability of emergence and one-year survival [60, 62]. Once this information is available, it could be easily included in our transition probability model. This version of our model will be able to inform if changes in their short-term mouse foraging decisions modulated by ungulate presence and predation risks have an imprint on seedling recruitment.

## Concluding remarks

Our mechanistic approach provides new insights about the joint effect of habitat structure, competition and risk on the foraging behavior by scatter hoarders and its potential consequences on acorn dispersal. In the presence of ungulates and when predation risks were high, mice acted as opportunistic foragers and concentrated their activities beneath tree canopies. As a result, predation rates decreased and larger acorns had a higher probability of survival (at least in the short term). These results suggest that inefficient foraging by mice can reduce acorn predation and may promote dispersal. Also, they highlight the importance of competition and risk as modulators of the spatial and temporal dynamism of oak-rodent interactions [2]. Finally, though future work is needed to estimate long-term cache survival and seedling establishment, our results support the view that the presence of the full set of acorn consumers and top predators can facilitate seed dispersal effectiveness in conditional mutualisms [2, 15, 51]. This may be particularly important in habitats like dehesas, which depend on scatter-hoarders to ensure their long-term sustainability [53, 63]

## Supporting information

**S1 Database.**
(XLSX)

**S1 File. Structure of models and priors.**
(DOCX)

**S2 File. Posterior predictive checks.**
(DOCX)

**S3 File. Specifications of transition probability model for acorn dispersal.**
(DOCX)

**S4 File. Code for the transition probability model.**
(DOCX)

**S5 File. Changes in vegetation structure in dehesas.**
(DOCX)

## Acknowledgments

D. López, M. Fernández and C. L. Alonso helped during fieldwork. D. López, B. Ramos and M. de Pablo pre-processed the video recordings, and D. Gallego, D. Valero, A. Velasco, C. J. González and E. Sánchez visualized the recordings noting seed choices. Authorities of the Cabañeros National Park provided the official permissions to carry out field experiments. J. España provided common genet scats.

## Author Contributions

**Conceptualization:** Mario Díaz.

**Data curation:** Jesús Sánchez-Dávila, Alvaro Navarro-Castilla.

**Formal analysis:** Teresa Morán-López, Mario Díaz.

**Funding acquisition:** Ignasi Torre, Isabel Barja, Mario Díaz.

**Investigation:** Ignasi Torre, Alvaro Navarro-Castilla, Mario Díaz.

**Methodology:** Teresa Morán-López, Jesús Sánchez-Dávila, Ignasi Torre, Alvaro Navarro-Castilla, Isabel Barja, Mario Díaz.

**Project administration:** Mario Díaz.

**Writing – original draft:** Teresa Morán-López, Mario Díaz.

**Writing – review & editing:** Teresa Morán-López, Jesús Sánchez-Dávila, Ignasi Torre, Alvaro Navarro-Castilla, Isabel Barja.

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
