## [Decision Letter · Decision Letter 0]

16 Dec 2021

PONE-D-21-34252Biological integrity enhances the qualitative effectiveness of conditional mice-oak mutualismsPLOS ONE

Dear Dr. Mario Díaz

Thank you for submitting your manuscript to PLOS ONE. After careful consideration, we feel that it has merit but does not fully meet PLOS ONE’s publication criteria as it currently stands. Therefore, we invite you to submit a revised version of the manuscript that addresses the points raised during the review process.

Three reviewers evaluated your manuscript and made constructive contributions, and I mostly agree with the reviewers. While they found valuable information and were positive, also showed concern on several major issues, so it requires MAJOR CHANGES for your manuscript does meet our criteria for publication. Especially relevant are reviewer 1's comments on the design's difficulty to assess whether mouse-oak interaction changes between predation and mutualism scenarios affected by possible competition or predation risk, which is at the heart of the manuscript. Reviewer 2 strongly highlights the need to substantially improve the graphical presentation of the results. 

Please submit your revised manuscript by Jan 30 2022 11:59PM . If you will need more time than this to complete your revisions, please reply to this message or contact the journal office at plosone@plos.org. Please include the following items when submitting your revised manuscript:A rebuttal letter that responds to each point raised by the academic editor and reviewer(s). You should upload this letter as a separate file labeled 'Response to Reviewers'.A marked-up copy of your manuscript that highlights changes made to the original version. You should upload this as a separate file labeled 'Revised Manuscript with Track Changes'.An unmarked version of your revised paper without tracked changes. You should upload this as a separate file labeled 'Manuscript'.

We look forward to receiving your revised manuscript.

Kind regards,

Pedro G. Blendinger, Dr

Academic Editor

PLOS ONE

Journal Requirements:

D. López, M. Fernández and C. L. Alonso helped during fieldwork. D. López, B. Ramos and M. de Pablo pre-processed the video recordings, and D. Gallego, D. Valero, A. Velasco, C. J. González and E. Sánchez visualized the recordings noting seed choices. Authorities of the Cabañeros National Park provided the official permissions to carry out field experiments. J. España provided common genet scats. This study is a contribution to the projects RISKDISP (CGL2009-08430) and VULGLO (CGL2010-22180-C03-03), funded by the Spanish Ministry of Economy, and REMEDINAL3-CM (S2013/MAE-2719), funded by the Autonomous Community of Madrid. We declare no conflict of interest.

We note that you have provided funding information. However, funding information should not appear in the Acknowledgments section or other areas of your manuscript. We will only publish funding information present in the Funding Statement section of the online submission form. 

MD. Grant number S2013/MAE-2719. REMEDINAL3-CM project, funded by the Autonomous Community of Madrid. NO

Reviewers' comments:

Reviewer's Responses to Questions

**Comments to the Author**

1. Is the manuscript technically sound, and do the data support the conclusions?

Reviewer #1: Partly

Reviewer #2: Partly

Reviewer #3: Yes

2. Has the statistical analysis been performed appropriately and rigorously? 

Reviewer #1: Yes

Reviewer #2: I Don't Know

Reviewer #3: Yes

3. Have the authors made all data underlying the findings in their manuscript fully available?

Reviewer #1: Yes

Reviewer #2: Yes

Reviewer #3: Yes

4. Is the manuscript presented in an intelligible fashion and written in standard English?

Reviewer #1: Yes

Reviewer #2: Yes

Reviewer #3: Yes

5. Review Comments to the Author

Reviewer #1: This study uses an experimental approach to investigate how the exclusion of ungulates and the presence of predator cues affects the removal, transportation, and short-term fate of Quercus ilex acorns handled by Algerian mice. The results show that the mice preferred large acorns for immediate consumption, but that this selectivity was changed by the presence of ungulates or predator cues, as well as the availability of more acorns. Acorns that were removed for caching were generally smaller, but this preference was changed in experiments with predator scents or ungulates present. Caching distances were generally very short and further reduced by the presence of full moon and outside of ungulate exclosures. Finally, the immediate post-dispersal predation of acorns was determined by acorn size, timing, and the presence of ungulates.

Based on their findings, the authors discuss how “environmental stress”, exerted by potential competition with ungulates and predation pressure, simulated by scents, moves the mouse-oak interaction from the predation end of the seed fate spectrum towards mutualism. This interpretation is hard to follow, based on the presented data, as the differences in “dispersal” distance (maybe a mean difference of 10cm) are biologically irrelevant and the time span over which post-dispersal predation is monitored does not suffice to make inferences about its effects on seed fate that leads to plant establishment. Therefore, I’d suggest to focus the discussion more on the decisions the mice face and how these are influenced in the experiment, rather than suggesting that this affects the whole ecosystem. Likely, the most important drivers are the presence of shrub cover, water availability/drought, and herbivory by large ungulates. Nonetheless, the experiment is interesting in teaching us about decisions that rodents make when handling seeds under different conditions. Therefore, I suggest the authors shift the discussion towards the behavioral ecology of the rodents, and away from effects of competition and predation on dispersal effectiveness.

L48: “seed dispersers” – otherwise the readers may think of natal or breeding dispersal of the hoarders themselves. Generally, please make sure to refer to “seed dispersal”, rather than simply “dispersal” (eg L65) for clarity

L48: To avoid the awkward term “acorn-bearing trees”, you could rephrase to “Scatter-hoarders are key seed dispersers in temperate and Mediterranean forests dominated by oaks [1-5]”

L56: “...space and time, making it difficult to predict...” (add comma and “it”)

L58: here and throughout: unless referring specifically to multiple individuals, please consider using the singular “mouse” rather than “mice” (e.g. “oak-mouse” in abstract etc).

L61: Please explain to the reader why competition encourages seed mobilization (just saw it on L66; consider moving that part forward a bit)

L58-72: While I understand the need to keep it short, it seems that the dynamics that turn a potential seed predation event into a seed dispersal event are oversimplified. Transportation distance alone does not make the interaction more antagonistic or mutualistic. While I know that not all aspects of seed fate can easily be quantified, it should at least be noted here that not only seed transport, but also consumption after caching can result in a seed predation event.

L123: please provide latin species name for the Algerian mouse

L127: do the mice live in or below the trees? (ie are they arboreal?) Consider rephrasing to “Mouse occupancy below target trees was established...”

L136: “consisted of...”

L151: For how many days did you search? This is really important for your definition of viability. Consider the fact that cached acorns may be retrieved and consumed weeks later.

L179: “acorn availability (g)” is this the natural crop of the tree or the overall mass of acorns provided to the mice during the experiment?

L182: scaling: please describe how and why

L183: “..intercept term to account for repeated sampling.”

Figures: Consider using “predator scent” to label the plots. Also, please add sample sizes to captions.

Figures 1 & 3: Does the color add any information? Otherwise consider making all figures black and white and differentiating yes/no with symbols or grayscale colors (eg Fig 3A)

L252: The last sentence reads as if it contradicts itself (more mobilization when no food, but more when lots of food). Consider rephrasing

L274: “environmental stress” is a very broad and loaded term, which could mean temperatures, etc. Please be concise

L280: “....probability to survive the first days after caching.” Since the whole point of caching is subsequent consumption, which may occur long after caching, I would be careful with this interpretation. However, by providing the time period over which you monitored seed fate, you can make this statement more accurate.

L282: Please expand on this notion of intermediate stress. The reference you provide in the introduction, Lichti et al. Biol Rev, only discusses the effect of intermediate seed availability on seed dispersal by rodents, but not overall stress. In the context of your study, you only compared the presence and absence of putative stressors, thus not providing quantitative support for the role of “intermediate stress”. Also, the argument that the recovery of cached seeds is affected slightly misses the point. Isn’t it about the moving of seeds in the first place? Why would the mouse take the risk to cache under high predation pressure, but then avoid that risk during recovery? In general, I think the “viability” argumentation is very limited by the time period over which the transported acorns were monitored (a few days, I assume).

L300: “Risk rather than competition modulates the effect of ungulate presence on acorn selection”. It seems to me that table 1 shows the opposite. The effect size of ungulate presence (size * ungulate) is nearly twice that of size*scent. Why wasn’t the interaction scent * ungulate included? It seems the experiment would be optimal to test the interaction between the two putative drivers of acorn selection.

L307: rephrase, hard to follow

Reviewer #2: General comment not to the Authors: "PLOS ONE does not copyedit accepted manuscripts, so the language in submitted articles must be clear, correct, and unambiguous. Any typographical or grammatical errors should be corrected at revision, so please note any specific errors here."

I have not proof-read the paper.

The study has a very interesting aim which is to explore whether biological integrity of the system can have positive effects on scatter hoarding and thus regeneration of the keystone oak. Results are generally nicely written-up, but figures need revisions. I have some comments regarding presentation, with the hope that these revisions can make the reading of the study easier.

Well explained experimental design – 40 trees, 10 assigned to a combination of herbivore exclusion crossed with genet scent addition.

Data analysis: sample size for the analysis needs to be provided, as it appears like model can be overfitted: 6 fixed effects + many interactions.

I am not familiar with Bayesian framework, so pardon the question. How the effects can be considered meaningful (or significant, but it is not about semantics) if their 95% CI overlap zero?

Figures needs major revisions. First, All figure captions should explain what is being showed, not provide the interpretation. The current version of Fig 1 is difficult to interpret, and the figure caption does not help as instead explaining what is shown, it presents the interpretation. Acorn size was categorial here with three levels, yet only two are shown. Then, if the acorn size was category, what is the point of showing it at y-axis which should be the place of response variable – I guess we are here mostily interested in probabilities? I suggest that all panels at Figure 1 show Probability at y-axis, acorns size at x-axis, and how the probability of each process changes with size depending on the treatment (ungulates, scent, moon etc).

Boxplots at Fig 2 should include data points in the background.

L50: The net outcome of the interaction does not depend on whether seeds are consumed or cached, as seeds are usually both consumed and cached in each interaction. The key is the balance between predation and dispersal, and the balance of the benefit (improved recruitment) vs cost (predation and thus reduced recruitment).

L58: This sentence suggests that corvids are inefficient dispersers in oaks savannas, yet we have papers showing the contrary both in dehesas (Baroja et al: https://besjournals.onlinelibrary.wiley.com/doi/abs/10.1111/1365-2745.13642) as well as in oak savannas in other regions (e.g. Pesendorfer et al, papers from California).

L70: “In the absence of stress… “ this sentence is oversimplification and sounds like rodents never store seeds in the absence of stress, which is not true. Simply put, if there is lots of food (no stress) we do expect that rodents will start to store.

L112: what does it mean that they were opened?

L121: averages of both areas are needed to support a statement that they have similar tree abundance

Reviewer #3: This is a comprehensive study as to the joint effect of habitat structure, competition and predation risk on dispersal effectiveness in an oak-mice system. They found that intermediate stress (presence of predator or grazer) could increase dispersal effectiveness and then facilitated interaction towards the mutualistic side, providing new evidences on conditional mutualism. I think this is a good contribution to the study of the field. I have only a few revision suggestions, mainly by including a few previous similar studies:

Line: 147-148. The wire-linked plastic seed tagging method was proposed by Xiao et al. (2006). You need to include the original reference: Xiao et al, 2006, Forest Ecology and Management, 223:18–23；

Line 296-299. You found ungulate presence would benefit dispersal of oak acorns. A previous study had similar results. You should include the reference in the discussion: Zhang et al. (2009), Wildlife Research, 36: 610–616;

6. PLOS authors have the option to publish the peer review history of their article (what does this mean?). If published, this will include your full peer review and any attached files.

Reviewer #1: No

Reviewer #2: No

Reviewer #3: No

---

## [Author Response · Author response to Decision Letter 0]

14 Feb 2022

Done

Done. No offcila permits were necessary, as we explained in the former version, now developed in lines 169-173 

D. López, M. Fernández and C. L. Alonso helped during fieldwork. D. López, B. Ramos and M. de Pablo pre-processed the video recordings, and D. Gallego, D. Valero, A. Velasco, C. J. González and E. Sánchez visualized the recordings noting seed choices. Authorities of the Cabañeros National Park provided the official permissions to carry out field experiments. J. España provided common genet scats. This study is a contribution to the projects RISKDISP (CGL2009-08430) and VULGLO (CGL2010-22180-C03-03), funded by the Spanish Ministry of Economy, and REMEDINAL3-CM (S2013/MAE-2719), funded by the Autonomous Community of Madrid. We declare no conflict of interest.

We note that you have provided funding information. However, funding information should not appear in the Acknowledgments section or other areas of your manuscript. We will only publish funding information present in the Funding Statement section of the online submission form. 

Done

MD. Grant number S2013/MAE-2719. REMEDINAL3-CM project, funded by the Autonomous Community of Madrid. NO

Done

---

## [Decision Letter · Decision Letter 1]

25 Apr 2022

PONE-D-21-34252R1Biological integrity of dehesa ecosystems favors acorn dispersal over predation in the mouse-oak mutualismPLOS ONE

Dear Dr. Mario Díaz

Thank you for submitting your manuscript to PLOS ONE. After careful consideration, we feel that it has merit but does not fully meet PLOS ONE’s publication criteria as it currently stands. Therefore, we invite you to submit a revised version of the manuscript that addresses the points raised during the review process.

ACADEMIC EDITOR:Thank you for submitting your revised manuscript. I apologize that this recommendation has taken longer than normal. It took us a long time to get reviewers for your revised manuscript, unfortunately the two reviewers of the first version were not available. All we agree that you have made many of the changes recommended by the former reviewers and the manuscript is much improved. However, both reviewers point out important details that should be taken into account. I direct you to the (new) reviews, where they pointed some important methodological and conceptual aspects that still remain unclear, as well as an over interpretation of some results that should be softened. Because I think there is potential in this manuscript, I am returning it to you for additional major revision. Please note that the concerns of the reviewers do need to be fully addressed.

We look forward to receiving your revised manuscript.

Kind regards,

Pedro G. Blendinger, PhD

Academic Editor

PLOS ONE

Reviewers' comments:

Reviewer's Responses to Questions

**Comments to the Author**

1. If the authors have adequately addressed your comments raised in a previous round of review and you feel that this manuscript is now acceptable for publication, you may indicate that here to bypass the “Comments to the Author” section, enter your conflict of interest statement in the “Confidential to Editor” section, and submit your "Accept" recommendation.

Reviewer #4: (No Response)

Reviewer #5: (No Response)

2. Is the manuscript technically sound, and do the data support the conclusions?

Reviewer #4: Partly

Reviewer #5: Yes

3. Has the statistical analysis been performed appropriately and rigorously? 

Reviewer #4: N/A

Reviewer #5: Yes

4. Have the authors made all data underlying the findings in their manuscript fully available?

Reviewer #4: Yes

Reviewer #5: Yes

5. Is the manuscript presented in an intelligible fashion and written in standard English?

Reviewer #4: Yes

Reviewer #5: Yes

6. Review Comments to the Author

Reviewer #4: The manuscript titled ‘Biological integrity of dehesa ecosystems favors acorn dispersal over predation in the mouse-oak mutualism’ explores the effects of predation risk cues and competition on seed size selection by rodents. The manuscript generally reads well but does not present clear novel idea. However, it combines few earlier ideas into more comprehensive study which presents complex interactions in dehesa system. Authors applied simple, commonly used both field and statistical methods presented in the Manuscript. The manuscript has been noticeably improved after previous review rounds. However, I still suggest a major revision before publication and provide some comments.

Title: I do not think that a term “biological integrity” reflects what has been presented in the study. Biological integrity represents the capability of supporting and maintaining a balanced, integrated, adaptive community of organisms having a species composition, diversity, and functional organization comparable to that of the natural habitat of the region before human alterations were imposed. I do not find such comparisons in the manuscript. If Authors still consider this term as somehow appropriate in the manuscript, please describe it more thoroughly in both introduction and discussion. If not, please change the title into more informative. Short title/Running headline:, i.e. “Predation and competition favor dispersal in mouse-oak interaction”, fits much better.

Authors use rarely used terms in seed dispersal studies: ‘select’, ‘mobilize’, ‘mobilization’, ‘transportation’ etc. What is the differences between “seeds selected” vs “seeds removed” (e.g. Fig 1)? Seeds selected include both those consumed in situ and those removed while seeds removed – only those taken away from the seed stations? I highly suggest to use commonly used terms, constistently throughout the text what makes interpretation much easier (not only less confusing but also increasing the probability of finding the article in searching results): ‘handle’/’handling’, ‘remove’/’disperse’, and ‘removal’/’dispersal’, etc. Otherwise, it creates a great confusion.

Line 122: It is not clear how many sites there are in total. Two with two subplots (one open and one exclosure) within each? Four separate plots?

Lines 129-130: Authors have mentioned that plant community structure is similar on both open and excluded areas. However, in Lines 315-316, they have assumed that obtained differences in rodent foraging can be caused by changes in vegetation cover and shrub layer. So, were there differences observed or not? If so, add some information about such changes, thus, I think ungulates may act functionally as vegetation changers rather than competitors in this system.

In the results, there is no information of:

- rodent diversity and abundances based on live-trapping and video recordings. Authors do not present any data confirming that seeds were handled by Algerian mice only while merely suggest that “The Algerian mouse (Mus spretus) is the most abundant scatter-hoarding rodent in the area [44] (…)”. This information seems to be crucial since seed:rodent ratio strongly affects rodent decisions. However, I assume Authors did not mark individuals (lines 169-171) so they can only provide information regarding rodent activity – while this not really reflects abundances and is a bit tricky: few bold individuals can be very active and they can provide a picture of higher abundances while shy individuals can be recorded or captured only once.

- numbers/percentages of acorns occurred in each category of seed removal process (seeds removed, consumed in situ, consumed after removal, cached etc.),

- three-way interactions, e.g. Size*Ungulates*Month. I am pretty sure that foraging activity (thus, the effect) of ungulates may depend on season.

I suggest to add these data to the results. Especially that, I suppose, these may differ between two seasons.

Lines 153-154: I wonder whether the results would be similar if Authors treated “seed size” as a numerical instead of categorical factor. Nature usually does not use such clear categories. Additionally, a term “seed size” is not suitable here as “size” refers to other measurements, such as length, width or volume, while Authors are focused on seed mass [g] instead. I suggest to change this nomenclature.

Line 193: What actually does ‘seed availability’ mean in this study? It is not explained in the text. Is it about naturally occurring acorns? Experimental acorns left in the seed stations? How was this measured?

Lines 203-204: Were distances measured for all seeds removed from seed stations (both consumed and cached)? “Seed dispersal distances” should be used and analyzed for cached seeds only, while “seed removal distances” can be analyzed for all seeds. If all seed distances were analyzed altogether ,“seed fate” should be added as a fixed factor to reveal differences in removal distances between cached and consumed seeds. Otherwise it is a bit confusing. Usually, consumed seeds are removed on shorter distances than cached so adding consumed seeds into analysis can alter the results.

Lines 252-253: The information regarding number of seeds as well as trees used should be provided in methods.

Lines 253-254: What about exclusures vs. controls? Was mouse activity altered by predator scent?

Lines 255-256: The diagram showing the whole seed fate process is clearly needed – this should include all numbers of seeds in all stages. Otherwise it is hard to follow throughout the results section. Please also add percentages to all numbers. What about missing seeds? Using seed-tagging method always provides some seeds that cannot be found.

Lines 271-272: Could removal distances vary between exclosures vs. controls (the effect of ungulates) due to differences in vegetation structure? For example, if ungulate activity leads to vegetation surrounding the focal trees (because it is normally grazed in open spaces) then removal distances will be shorter (because vegetation is closer to focal trees in open spaces/controls while more scattered in exclosures). Moreover, did Authors check for microhabitats chosen for caching in exclosures vs. controls?

Line 288: Please change ‘relaxed’ into more appropriate terminology, such as ‘not exposed to stressful factors’. I assume mice can never be truly relaxed.

Reviewer #5: The combination of including predation risk with ungulate exclosures in a disturbed landscape makes for an interesting study with novel results. This research provides important insights regarding rodent foraging decisions in anthropogenic habitats which can inform oak woodland restoration.

Overall:

The use of the words “mobilized” and “mobilization” instead of “dispersed” and “dispersal” (especially when “dispersal” is used in the title) is confusing. Maybe the authors don’t consider movement away from the parent plant “dispersal” unless seedling recruitment is the final fate? Whatever the reason, there is a whole body of dispersal literature that uses the term “dispersed” or “dispersal” to describe seed movement away from the focal plant. Also, due to the definition of mobilization, the term can carry with it a military connotation and doesn’t seem to be the best term to replace “dispersal” if that was the authors’ intent (mobilization: 1. the action of a country or its government preparing and organizing troops for active service; 2. the action of making something movable or capable of movement). Trying to make the manuscript less repetitive by using “mobilization” or “movement” every so often is understandable, but it is overused in the manuscript and shouldn’t be used along with “distance” since “dispersal distance” is a pretty well-established term. The seed dispersal literature is a quite large body of work at this point with its own established terminology that creates continuity, so it’s confusing and unnecessary to use new terminology without any justification.

The Methods and Results are lacking some important details for a study with “dispersal” in the title. How far from the cages did the search area for dispersed acorns extend? What was considered “predated”? Were acorn fragments found or tags left behind? Since dispersed acorns were tracked “throughout the experiment,” was any secondary dispersal of cached acorns documented? If tagged acorns were removed and never found, could they not have been dispersed outside of the search area? 21% of removed acorns were not relocated but nothing is mentioned in the Results or Discussion about their potential fate. How many of those that were dispersed and recovered were cached/buried? “Scatter-hoarding” is the first word in the abstract, yet there is very little information about the “scatter-hoards” that were recovered in the Results. Cache depth and microsite placement are important components of effective seed dispersal but are not touched on at all. In terms of microsite deposition, were all 211 relocated acorns found under focal tree canopies? Were any moved to the canopies of neighboring trees or into the open?

In the Discussion, lines 344-348, the authors overstated the completeness of their study in terms of seed dispersal effectiveness. The study did not examine cache microsites and seedling recruitment where seeds are deposited (qualitative component), nor did they report details like number of visits or number if seeds per visit (quantitative) (Schupp et al. 2010).

Suggestions for minor edits are highlighted and written in red text in the attached PDF.

7. PLOS authors have the option to publish the peer review history of their article (what does this mean?). If published, this will include your full peer review and any attached files.

Reviewer #4: No

Reviewer #5: No

---

## [Author Response · Author response to Decision Letter 1]

21 Jun 2022

ACADEMIC EDITOR:

Thank you for submitting your revised manuscript. I apologize that this recommendation has taken longer than normal. It took us a long time to get reviewers for your revised manuscript, unfortunately the two reviewers of the first version were not available. All we agree that you have made many of the changes recommended by the former reviewers and the manuscript is much improved. However, both reviewers point out important details that

should be taken into account. I direct you to the (new) reviews, where they pointed some important methodological and conceptual aspects that still remain unclear, as well as an over interpretation of some results that should be softened. Because I think there is potential in this manuscript, I am returning it to you for additional major revision. Please

note that the concerns of the reviewers do need to be fully addressed.

Dear editor, thank you for providing us the opportunity to revise this work. Concerns raised by the new reviewers have been very helpful at clarifying some aspects of our methods and results. For instance, that the main effect of ungulate presence is mediated by changes in vegetation cover rather than on competition per se. In addition, comments have been very constructive so that our discussion better reflects our results. We believe that the current version has improved the first revision. Please see specific comments below.

COMMENTS FROM REVIEWERS

Reviewer #4: The manuscript titled ‘Biological integrity of dehesa ecosystems favors acorn dispersal over predation in the mouse-oak mutualism’ explores the effects of predation risk cues and competition on seed size selection by rodents. The manuscript generally reads well but does not present clear novel idea. However, it combines few earlier ideas

into more comprehensive study which presents complex interactions in dehesa system. Authors applied simple, commonly used both field and statistical methods presented in the Manuscript. The manuscript has been noticeably improved after previous review rounds. However, I still suggest a major revision before publication and provide some comments.

We appreciate the support to our work and the constructive comments that have helped us to be more precise and clearer about our results.

Title: I do not think that a term “biological integrity” reflects what has been presented in the study. Biological integrity represents the capability of supporting and maintaining a balanced, integrated, adaptive community of organisms having a species composition, diversity, and functional organization comparable to that of the natural habitat of the

region before human alterations were imposed. I do not find such comparisons in the manuscript. If Authors still consider this term as somehow appropriate in the manuscript, please describe it more thoroughly in both introduction and discussion. If not, please change the title into more informative. Short title/Running headline:, i.e. “Predation and

competition favor dispersal in mouse-oak interaction”, fits much better.

We appreciate this comment since the previous title could lead to confusions for readers. We have changed it to “Ungulate presence and predation risks reduce acorn predation by mice in dehesas”. We have slightly changed the title because in the case of ungulates our interpretation is that their main effect is mediated by changes in the vegetation cover rather than competition per se. Also, since we have decided to focus on acorn predation in the title since it is more precise with respect to our findings (we did not quantify seed dispersal effectiveness).

Authors use rarely used terms in seed dispersal studies: ‘select’, ‘mobilize’, ‘mobilization’, ‘transportation’ etc. What is the differences between “seeds selected” vs “seeds removed” (e.g. Fig 1)? Seeds selected include both those consumed in situ and those removed while seeds removed – only those taken away from the seed stations? I highly suggest to use commonly used terms, constistently throughout the text what makes interpretation much easier (not only less confusing but also increasing the probability of finding the article in searching results): ‘handle’/’handling’, ‘remove’/’disperse’, and ‘removal’/’dispersal’, etc. Otherwise, it creates a great confusion.

To define the stages of the foraging decisions processes we followed that of Wang et al. 2013 (Oikos). That is (1) when encountering acorns which one to handle (acorn selection); (2) whether to remove the selected acorn away or not, (3) how far to carry it and (4) whether to predate it once deposited or not (L185-188). Selection is the first foraging decision made by rodents when encountering acorn. It is a term frequently used in literature and better reflects the ecological process we are evaluating (i.e. which acorn to handle among those available). In fact, it is modeled as a multinomial regression. Therefore, we have decided to continue calling the first decision as “acorn selection” but refer to “handled acorns” throughout the text (e.g. Fig 1, L186, L192). In the case of removal, we were referring to the decision of removing the acorn outside the cage or not (L184 and 201, in the former version of the manuscript). However, it may have not been clear enough. We have highlighted this throughout the text and in figure 1 (e.g. Fig. 1, L201). Regarding the use of term dispersal, we decided to refer to transportation rather than dispersal because a high proportion of seeds that are taken away are finally predated.

Line 122: It is not clear how many sites there are in total. Two with two subplots (one open and one exclosure) within each? Four separate plots?

In the study area there are two sites and each one has an exclosure and a “control” plot (with ungulates). To evaluate the effects of ungulate presence we established at each site half of focal trees inside the exclosure and half outside it. We have rewritten this section to make it clearer (L118-132).

Lines 129-130: Authors have mentioned that plant community structure is similar on both open and excluded areas. However, in Lines 315-316, they have assumed that obtained differences in rodent foraging can be caused by changes in vegetation cover and shrub layer. So, were there differences observed or not? If so, add some information about such changes, thus, I think ungulates may act functionally as vegetation changers rather than

competitors in this system.

This is indeed an important issue that was not clear enough in the previous version of the manuscript. In general, the study area shows a low tree density and shrub cover (1%). It is a savanna-like environment where understory cover is mainly composed by grasses and tussocks. Lack of browsing by ungulates inside exclosures has resulted in taller resprouts under canopies and vegetation around trees (mainly herbs and tussocks). Such changes mediate the availability of antipredatory cover for mice. Information about differences in vegetation cover outside and inside exclosures has been clarified in the material and methods section (L120-122, L127-132, FigS5_1).

In the results, there is no information of:

- rodent diversity and abundances based on live-trapping and video recordings. Authors do not present any data confirming that seeds were handled by Algerian mice only while merely suggest that “The Algerian mouse (Mus spretus) is the most abundant scatter-hoarding rodent in the area [44] (…)”. This information seems to be crucial since seed:rodent ratio strongly affects rodent decisions. However, I assume Authors did not

mark individuals (lines 169-171) so they can only provide information regarding rodent activity – while this not really reflects abundances and is a bit tricky: few bold individuals can be very active and they can provide a picture of higher abundances while shy individuals can be recorded or captured only once.

 In L138-140 we specified that to select focal trees we live-trapped mice and captured Mus spretus with a high detection probability. Also, we could identify the species in the video recording of foraging events. These issues have been clarified in L254-256, 258. We agree that without marking individuals or adjusting occupancy-detection models we are actually inferring a “naïve occupancy”. Nonetheless, in our study area the probability of detection of rodents under tree canopies is high (0.88±0.03) and we live-trapped mice during two consecutive nights (L138-140). Thus, we believe that differences between “naïve” and actual occupancies should be low. We also agree that seed:rodent ratios are important modulators of mouse foraging decisions. However, in our study, main differences in such ratios in November and February are driven by acorn availability (during and outside the acorn fall peak). In fact, no acorns were found beneath tree canopies in February (L254). 

In sum, some information was already available in the previous version of the manuscript but it may have not been sufficiently highlighted. We have tried to clarify these issues. In addition, given the high detection probabilities in our area and large differences in acorn availability between months, we believe assumptions made by our approach are reasonable.

- numbers/percentages of acorns occurred in each category of seed removal process (seeds removed, consumed in situ, consumed after removal, cached etc.),

We have completed this information in L255-258.

- three-way interactions, e.g. Size*Ungulates*Month. I am pretty sure that foraging activity (thus, the effect) of ungulates may depend on season. I suggest to add these data to the results. Especially that, I suppose, these may differ between two seasons.

We agree that evaluating changes in the effects of ungulates between months is interesting. Also, other three-way interactions like moonlight, risk and size. We have been suggested multiple three-way interactions in previous revisions of this manuscript. Nonetheless, we have decided to keep our analyses as simple as possible for two main reasons. First, the aim of our study is to assess the “average” effect of competition and risk on size-driven foraging decisions by mice. To account for changes on “baseline” brightness and acorn availability, we performed our experiment under new and full moon conditions as well as in February and November (L169-172). We did not intend to analyze all factors and their possible interactions but to better understand the effects of acorn size, competition and risk. In this sense, our current analyses reflect our specific aims. The second reason is related to sample sizes. Even though we tracked a large number of acorns >1000, at stages of acorn transportation and initial fates our sample size is much lower, making it very difficult to obtain reliable estimates of three-way interactions. Nonetheless, we have tried to fit the model for acorn selection and removal following your suggestions. When fitting the multinomial regression (i.e. acorn section) we had a lot of problems for model convergence in the variance of the random term. This warned us that adjusting such complex models may not be possible with information in our data and design, as suspected. Therefore, given that our current analyses better reflect our specific aims and that our data does not allow to obtain reliable estimates of three-way interactions for all foraging decisions, we have kept the analyses as it was in the former version.

Lines 153-154: I wonder whether the results would be similar if Authors treated “seed size” as a numerical instead of categorical factor. Nature usually does not use such clear categories. Additionally, a term “seed size” is not suitable here as “size” refers to other

measurements, such as length, width or volume, while Authors are focused on

seed mass [g] instead. I suggest to change this nomenclature.

As stated in the previous version, we treated size as numerical in our analyses (now L193). However, the way we explained the experimental set-up may have been confusing. To have a full range of acorn sizes that were balanced, we selected 5 small, 5 medium and 5 large acorns. This has been rewritten 150-153 to avoid any confusion. The term acorn size is frequently used in cafeteria experiments that evaluate rodent foraging decisions. We have decided to keep the same term but we have specified more clearly that we are referring to grams (L193). 

Line 193: What actually does ‘seed availability’ mean in this study? It is not explained in the text. Is it about naturally occurring acorns? Experimental acorns left in the seed stations? How was this measured?

Local acorn availability was measured as total acorn mass in the cage during the event. This information was in the previous version (L192-193, now L195-196).

Lines 203-204: Were distances measured for all seeds removed from seed stations (both consumed and cached)? “Seed dispersal distances” should be used and analyzed for cached seeds only, while “seed removal distances” can be analyzed for all seeds. If all seed distances were analyzed altogether ,“seed fate” should be added as a fixed factor to

reveal differences in removal distances between cached and consumed seeds. Otherwise it is a bit confusing. Usually, consumed seeds are removed on shorter distances than cached so adding consumed seeds into analysis can alter the results.

In general, when evaluating the so-called “dispersal” kernels of acorn mobilization by rodents all removed acorns are taken into account. Even though this is not measuring dispersal it provides very important information about the ability of rodents to move throughout the landscape and about their decision of “how far to carry the seed”. Since predation rates by rodents tend to be high, there is usually not enough sample size to estimate dispersal kernels on solely cached acorns, especially in managed systems like dehesas. We have used the term acorn transportation throughout the text (instead of dispersal) because we are aware that removal and dispersal are not synonyms in the case of mouse-acorn interactions. We have rewritten this sentence to clarify it L205-207.

We agree that frequently predated seeds are dispersed shorter distances. However, including seed fate as a fixed factor in the analyses of transportation distances would mix two foraging decisions (how far to carry a seed and whether to consume it or not). This would be a little bit confusing with respect to the aims of our work, which is evaluating the effects of acorn size and environmental variables on each foraging decision (selection, removal, transportation and predation). In any case, we have performed the analyses as suggested and there was no difference in mobilization distances mediated by seed fate (mean effect -0.21 [-0.27, 0.43]). Therefore, we have decided to continue with the current analyses. 

Lines 252-253: The information regarding number of seeds as well as trees used should be provided in methods.

Information about the number of acorns tracked was a result since not all focal trees had mice activity. This was not clear enough so we have rewritten this part of the text (L256-261).

Lines 253-254: What about exclusures vs. controls? Was mouse activity altered by predator scent?

We specified new and full moon conditions in both months because these were the two factors defining our cafeteria experiment set-up (L167-169). Therefore, the combination of moon and month was relevant to specify the level of activity in our focal trees, and hence, the number of acorns tracked, removed etc. Probably it was not clear enough in the previous version of the manuscript. This part has been rewritten. 

We have calculated the number of trees with activity outside and inside exclosures and also in trees with and without predator scent. We provide it below but we believe it is not relevant for our work.

Table S1. Number of trees showing mouse activity (in different treatments.

Month Moon Ungulate presence Predator scent

November New (18) No (11) No (9)

 Yes (7) Yes (9)

 Full (26) No (13) No (11)

 Yes (13) Yes (15)

February New (26) No (13) No (10)

 Yes (13) Yes (16)

 Full (24) No (11) No (11)

 Yes (13) Yes (15)

Lines 255-256: The diagram showing the whole seed fate process is clearly

needed – this should include all numbers of seeds in all stages. Otherwise it is hard to follow throughout the results section. Please also add percentages to all numbers. What about missing seeds? Using seed-tagging method always provides some seeds that cannot be found.

Some of the information required like missing seeds was in the previous version of the manuscript. However, we did not provide enough information about the percentage and number of acorns throughout all decisions made by rodents. It has been completed as suggested in L256-261. We do not include a diagram because we would have needed to split different months, areas with and without ungulates, moon-light conditions… We believe that with the information now completed in the main text and our analyses we provide a full picture of our results.

Lines 271-272: Could removal distances vary between exclosures vs. controls (the effect of ungulates) due to differences in vegetation structure? For example, if ungulate activity leads to vegetation surrounding the focal trees (because it is normally grazed in open spaces) then removal distances will be shorter (because vegetation is closer to focal trees in open spaces/controls while more scattered in exclosures). Moreover, did Authors

check for microhabitats chosen for caching in exclosures vs. controls?

Indeed, the effects of ungulate presence may be related to change in vegetation structure rather than to competition (as discussed in the previous draft). Inside exclosures, vegetation around focal trees was higher and also oak resprouts. Accordingly, mice were able to perform an acorn selection behaviour and mobilized seeds further. We have rewritten some parts of the discussion to stress these ideas see L316-320, 322-324, 361-363.

Regarding microhabitats, we recorded the type of microhabitat where acorns were left after transportation. Most of acorns were deposited under tree canopies or resprouts (96.8 and 97.9% of them, inside and outside exclosures). This information is now provided L280-283.

Line 288: Please change ‘relaxed’ into more appropriate terminology, such as ‘not exposed to stressful factors’. I assume mice can never be truly relaxed.

Changed accordingly.

Reviewer #5: The combination of including predation risk with ungulate exclosures in a disturbed landscape makes for an interesting study with novel results. This research provides important insights regarding rodent foraging decisions in anthropogenic habitats which can inform oak woodland restoration.

We are grateful for the support to our work and constructive comments. We have tried to be clearer with respect to some terms used throughout the text. Also, we have toned down some parts of the discussion according to the concerns raised throughout the revision.

Overall:

The use of the words “mobilized” and “mobilization” instead of “dispersed” and “dispersal” (especially when “dispersal” is used in the title) is confusing. Maybe the authors don’t consider movement away from the parent plant “dispersal” unless seedling

recruitment is the final fate? Whatever the reason, there is a whole body of dispersal literature that uses the term “dispersed” or “dispersal” to describe seed movement away from the focal plant. Also, due to the definition of mobilization, the term can carry with it a

military connotation and doesn’t seem to be the best term to replace “dispersal” if that was the authors’ intent (mobilization: 1. The action of a country or its government preparing and organizing troops for active service; 2. the action of making something movable or capable of movement). Trying to make the manuscript less repetitive by using

“mobilization” or “movement” every so often is understandable, but it is overused in the manuscript and shouldn’t be used along with “distance” since “dispersal distance” is a pretty well-established term. The seed dispersal literature is a quite large body of work at this point with its own established terminology that creates continuity, so it’s confusing and unnecessary to use new terminology without any justification.

We agree that the term dispersal has been widely used for scatter-hoarding rodents. Nonetheless, from our perspective using this term is confusing because frequently rodents (and specially mice) consume seeds that have been carried away. In such cases they act as seed predators rather than dispersers. Therefore, coining the term dispersal to transported seeds may be confusing with respect to the actual role of mice. This is the main reason why we used the term “mobilization” throughout the text. We were not aware that “mobilization” could be confounded with military terms (English is not our mother tongue). Therefore, we have removed mobilization and have used “transportation” instead.

The Methods and Results are lacking some important details for a study with “dispersal” in the title. How far from the cages did the search area for dispersed acorns extend? 

We searched for acorns within a 30 m-radius circle around focal trees (and cages). This represents an area where most acorns are deposited by mice (at least in the short term) and, in principle, a high proportion of Mus spretus home ranges (see comments below). Information now available in L160-162.

What was considered “predated”? fragments found or tags left behind? Since dispersed acorns were tracked “throughout the experiment,” was any secondary dispersal of cached

acorns documented?

Following Perea et al.’s (2011) experiments, we considered an acorn to be predated when it was completely consumed or partially consumed with embryo damage. Now in L164-167.

No secondary acorn dispersal event was observed in our experiments, as well as during previous work with this species in the same area (Muñoz & Bonal 2007, 2011); this behavior seems rare even for wood mice, a much more proficient acorn disperser owing to its larger size (Perea et al. 2011). 

If tagged acorns were removed and never found, could they not have been dispersed outside of the search area? 21% of removed acorns were not relocated but nothing is mentioned in the Results or Discussion about their potential fate.

Unfound acorns may have been mobilized further than 30 m or predated by other acorn consumers characterized by larger home ranges than mice (e.g. ungulates). We searched for acorns within a quite large area around focal trees, where most acorns are usually deposited and compromising an area adequate for Mus spretus home ranges (e.g. Gray et al. 1998). Therefore, we believe that our findings reflect general patterns of acorn removal distances and predation rates. We prefer to not speculate about unfound acorns because their fate is very uncertain.

Gray, S. J. et al. 1998. Microhabitat and spatial dispersion of the grassland mouse (Mus spretus Lataste). – J. Zool. 246: 299–308.

How many of those that were dispersed and recovered were cached/buried? “Scatter-Hoarding” is the first word in the abstract, yet there is very little information about the

“scatter-hoards” that were recovered in the Results. Cache depth and microsite placement are important components of effective seed dispersal but are not touched on at all. 

We now explicitly state that only 8 acorns (11.9% of seeds dispersed and not predated) were found buried (i.e. cached) L260.

In terms of microsite deposition, were all 211 relocated acorns found under focal tree canopies? Were any moved to the canopies of neighboring trees or into the open?

Most acorns were found beneath tree canopies or resprouts (now L280-283). Since a low proportion of acorns were found in open areas (2.37%) the actual probability of recruitment of transported seeds was very low. We have toned down some parts of the discussion accordingly. L349-353.

In the Discussion, lines 344-348, the authors overstated the completeness of their study in terms of seed dispersal effectiveness. The study did not examine cache microsites and seedling recruitment where seeds are deposited (qualitative component), nor did they report details like number of visits or number if seeds per visit (quantitative) (Schupp et al. 2010).

We agree that in the previous version the completeness of our study was overstated. In fact, low caching rates (<1% of the seeds removed) did not allow us to evaluate the effects of environmental factors on actual scatter-hoarding by mice. We are confident that the present work provides new information about how different environmental factors affect mouse foraging decisions in dehesas, and hence, potential recruitment. This is quite important since corvids are usually lacking when dehesas do not have nearby forests. Nonetheless, we agree that some parts of the discussion were too optimistic about our approach. Therefore, we have rewritten them and hopefully now they better reflect the strengths and limitations of our work. L358-382.

Suggestions for minor edits are highlighted and written in red text in the

attached PDF.

Thank you for the throughout revision of the text. It has been very helpful to correct some grammar mistakes and clarify some sentences.

L85. In the sentence: “Nonetheless, the strength and even sign of acorn size effects on mouse foraging decisions are not unequivocal, but context-dependent”. Commented: “What’s the sign of acorn size?”. 

In this sentence strength and sign are referring to the effects of acorn size. They can be positive or negative (sign) and have a different absolute value (strength).

---

## [Editor Report · Decision Letter 2]

11 Jul 2022

Ungulate presence and predation risks reduce acorn predation by mice in dehesas

PONE-D-21-34252R2

Dear Dr. Díaz,

We are pleased to inform you that your manuscript has been judged scientifically suitable for publication and will be formally accepted for publication once it meets all outstanding technical requirements.

Within one week, you will receive an e-mail detailing the required amendments. When these have been addressed, you will receive a formal acceptance letter and your manuscript will be scheduled for publication.

Kind regards,

Pedro G. Blendinger, PhD

Academic Editor

PLOS ONE

---

## [Editor Report · Acceptance letter]

1 Aug 2022

PONE-D-21-34252R2 

Ungulate presence and predation risks reduce acorn predation by mice in dehesas 

Dear Dr. Díaz:

I'm pleased to inform you that your manuscript has been deemed suitable for publication in PLOS ONE. Congratulations! Your manuscript is now with our production department. 

Kind regards, 

on behalf of

Dr. Pedro G. Blendinger 

Academic Editor

PLOS ONE